# Multifaceted Regulation of MicroRNA Biogenesis: Essential Roles and Functional Integration in Neuronal and Glial Development

**DOI:** 10.3390/ijms22136765

**Published:** 2021-06-23

**Authors:** Izabela Suster, Yue Feng

**Affiliations:** Department of Pharmacology and Chemical Biology, Emory University School of Medicine, Atlanta, GA 30322, USA; isuster@emory.edu

**Keywords:** microRNA biogenesis, neuronal differentiation, glial lineage, brain development

## Abstract

MicroRNAs (miRNAs) are small, non-coding RNAs that function as endogenous gene silencers. Soon after the discovery of miRNAs, a subset of brain-enriched and brain-specific miRNAs were identified and significant advancements were made in delineating miRNA function in brain development. However, understanding the molecular mechanisms that regulate miRNA biogenesis in normal and diseased brains has become a prevailing challenge. Besides transcriptional regulation of miRNA host genes, miRNA processing intermediates are subjected to multifaceted regulation by canonical miRNA processing enzymes, RNA binding proteins (RBPs) and epitranscriptomic modifications. Further still, miRNA activity can be regulated by the sponging activity of other non-coding RNA classes, namely circular RNAs (circRNAs) and long non-coding RNAs (lncRNAs). Differential abundance of these factors in neuronal and glial lineages partly underlies the spatiotemporal expression and function of lineage-specific miRNAs. Here, we review the continuously evolving understanding of the regulation of neuronal and glial miRNA biogenesis at the transcriptional and posttranscriptional levels and the cooperativity of miRNA species in targeting key mRNAs to drive lineage-specific development. In addition, we review dysregulation of neuronal and glial miRNAs and the detrimental impacts which contribute to developmental brain disorders.

## 1. Introduction

MicroRNAs (miRNAs) are a class of small, non-coding RNAs (20–24 nt) that suppress mRNA targets through mRNA degradation and translational repression [1]. Mature miRNAs are highly conserved across species and broadly expressed but display variable tissue-specific abundance, with a subset of miRNAs enriched in or specific to the brain [2,3]. As such, the expression profiles and function of brain miRNAs in normal neuronal development have been the focus of numerous investigators in past decades. From the resultant studies, neuronal-specific/brain-enriched miRNAs have been demonstrated to govern all stages of neuronal lineage development, from proliferation of neural progenitor cells (NPCs), neuronal fate commitment, morphological differentiation, circuitry assembly to synaptic plasticity [4,5,6,7]. In addition, emerging evidence has also revealed essential roles of miRNAs in glial development and function, which can impact neurons through cell non-autonomous mechanisms via glia-neuron interactions [8,9,10]. Importantly, dysregulation of miRNAs in neurons and glia has been found in various human brain developmental diseases, including schizophrenia, autism spectrum disorders (ASD) and intellectual disability [11,12,13]. Moreover, investigation of NPCs and neurons derived from human induced pluripotent stem cells (iPSCs) suggests that miRNA malfunctioning in early neuronal development may contribute to the pathogenesis of various age-related neurodegenerative diseases [14,15].

Despite the large volume of discoveries that document the essential roles of miRNAs in brain development and function, molecular mechanisms that precisely regulate miRNA biogenesis in neuronal and glial development and thus impact normal and diseased brains have just begun to unfold. How distinct miRNAs converge on downstream molecular networks to advance neuronal development is a prevailing research question under active investigation. Given the increasingly appreciated functional importance of glia in modulating neuronal function, uncovering the role of glia-expressed miRNAs in neuron–glia interaction/communication has become a frontier field of research.

Here we provide an up-to-date and comprehensive review of the recent advancements and outstanding questions regarding mechanisms that control miRNA biogenesis and functional abundance in neuronal and glial lineage development, as well as how diverse neural miRNAs converge on common downstream molecular networks. In addition, we highlight recent discoveries regarding important roles of miRNAs in neuron–glia communication, malfunction of miRNAs in neurodevelopmental diseases and therapeutic potential of miRNAs in various brain disorders. 

## 2. Canonical and Non-Canonical miRNA Processing Pathways

### 2.1. The Canonical miRNA Biogenesis Pathway

Decades of investigation have established a canonical miRNA biogenesis pathway beginning with the transcription of a miRNA host gene by RNA polymerase II (RNAPII) to produce a primary miRNA (pri-miRNA) transcript [16]. Pri-miRNAs contain a local stem-loop structure that encodes miRNA duplexes in the arm of the stem. Cleavage of the stem-loop by the RNase III enzyme Drosha and its RNA-binding protein cofactor DiGeorge Syndrome Critical Region 8 (DGCR8), together termed the Microprocessor complex, produces a 60–80 nt stem-loop/hairpin intermediate known as the precursor miRNA (pre-miRNA) [17,18]. Nuclear export of the pre-miRNA is facilitated by Exportin 5 and Ran-GTP [19]. In the cytoplasm, a second RNase III enzyme, Dicer, cleaves the pre-miRNA terminal loop to produce a miRNA duplex [20,21]. Each miRNA duplex produces two mature miRNAs: one from the 5′ stand and one from the 3′ strand. Using miR-124 as an example, these miRNAs are termed miR-124-5p and miR-124-3p, respectively. Generally, one of the arms/strands of the duplex, namely the guide strand, is preferentially loaded onto Argonaute (AGO) to form the RNA-induced silencing complex (RISC), which acts as an endogenous gene silencer directed by miRNA base-pair complementarity with the 3′ untranslated region (UTR) of target mRNAs [22,23]. The other strand of the miRNA duplex, the passenger strand, denoted as miRNA*, is often found in much lower abundance [24]. Nonetheless, mutually exclusive expression of guide and passenger strands is rarely observed [25]. For certain miRNA species, the passenger strand is not subjected to rapid degradation but rather, may acutely target mRNAs and play functionally important roles. A notable example is miR-9-3p (miR-9*), which targets anti-neural transcriptional machinery [26,27]. 

Mounting evidence has demonstrated the crucial roles of the canonical miRNA biogenesis enzymes in proper brain development and function. Conditional knockout of *Drosha* in mouse NPCs resulted in a loss of multipotency status and induced precocious neuronal differentiation, which was phenocopied by depletion of DGCR8 [28]. Dicer depletion profoundly impairs the morphological and proliferative characteristics of NPCs [29], mature neurons [30,31,32], oligodendroglia [33], microglia [34] and astrocytic glia [35,36]. Developmental regulation of canonical miRNA biogenesis enzymes in neurons has not been precisely characterized. However, the loss of miR-107 resulted in aberrantly increased expression of Dicer and miR-9, which in turn led to excessive neurogenesis during zebrafish hindbrain development, suggesting that miR-107 is a modulator of Dicer to maintain homeostatic levels of pro-neurogenic miRNAs [37]. In mature neurons, Drosha is restricted in the soma while Dicer and RISC components are found in distal dendrites and axons [38,39], potentially allowing for spatially-restricted maturation of miRNAs. Indeed, a recent study demonstrated biogenesis of mature miRNAs from fluorescently labeled pre-miRNAs in neuronal dendrites upon synaptic stimulation, which reduced local protein synthesis of predicted mRNA targets [40]. Thus, despite the ubiquitous expression of the canonical miRNA biogenesis machinery, neurons harbor more sophisticated spatial regulation of canonical miRNA biogenesis that governs the development and function of the normal brain.

### 2.2. Emerging Roles of Non-Canonical miRNA Biogenesis Pathways in the Central Nervous System

Alternative non-canonical pathways for miRNA biogenesis exist, which can bypass steps of the aforementioned canonical pathway. Mechanisms for both Drosha-independent and Dicer-independent miRNA biogenesis have been reviewed by Yang and Lai [41]. Mirtrons, derived from splicing of short introns that carry pre-miRNA-like structures, are recognized and processed by Dicer, constituting one group of Drosha-independent non-canonical miRNAs. Of note, a number of mirtron-derived miRNAs are highly expressed in the mouse hippocampus and cerebral cortex, including miR-877 and miR-1981 [42]. On the other hand, Drosha/DGCR8-dependent but Dicer-independent miRNA biogenesis has also been observed, represented by miR-451 [43,44,45]. Mechanistically, the endonucleolytic slicer activity of Ago2 cleaves the pre-miR-451 hairpin into a ∼30 nt intermediate RNA in the cytoplasm, which is trimmed into a conventional ∼22 nt miRNA by 3′-exonucleolytic activity of the poly(A)-specific ribonuclease (PARN) [45]. Notably, miR-451 was reported to drive glioma tumorigenesis [46,47]. Moreover, miR-451 exerts neuroprotective effects against cerebral ischemia/reperfusion injury in stroke patients [48]. These emerging studies suggest potential roles of non-canonical miRNA-biogenesis pathways in normal and diseased brains. Lastly, another class of Dicer-independent miRNA-like molecules are derived from Argonaute-associated short introns of 80–100 nucleotides, termed agotrons, which are stabilized by AGO proteins and capable of repressing mRNAs via sequence seed-matching in the 3′-UTR of the targets [49]. 

## 3. MiRNAs Play Key Roles in Governing Neuronal and Glial Development

### 3.1. The Most Abundant MiRNA in the Brain: MiR-124 and its Anti-Neurogenic Targets 

The mature miR-124 (miR-124-3p) is one of the most well-characterized pro-neurogenic miRNAs, which governs various aspects of neuronal development and function, including neurogenesis, neuronal network assembly and synaptic plasticity [50,51,52,53]. The sequence of miR-124 is highly conserved from worms to humans. Brain-specific expression of miR-124 was identified in the first large-scale vertebrate miRNA expression profiling panel [2]. Brain-specific expression of miR-124 in humans has also been observed [3]. During neuronal maturation, miR-124 is minimally expressed in NPCs and immature neurons but is drastically up-regulated to become the most abundantly expressed miRNA in the adult murine brain, accounting for an estimated 25% to 48% of all brain miRNAs [2,54,55]. Expression of miR-124 has been detected as early as E11.5 in the central nervous system (CNS) of mouse embryos [55]. In contrast to the functional miR-9* mentioned above, miR-124-5p (miR-124*) is not appreciably expressed in the embryonic CNS, suggesting that miR-124* does not play a significant role in prenatal neurodevelopment [55]. Analysis of a miR-124 activity sensor transgenic mouse model at E13.5 and in the adult brain supported CNS neuron-specific miR-124 activity, with no activity detected in astrocytes, microglia and the peripheral nervous system (PNS) [56]. 

The abundant pool of miR-124 generated in the CNS targets numerous well-defined repressors of neuronal differentiation, including but not limited to RhoG, PAX3 and BAF53a [27,57,58]. Of note, miR-124 has also been reported to target the anti-neural epigenetic regulator histone methyltransferase EZH2. Importantly, miR-124 targets master repressors of neuronal-specific gene expression. More specifically, miR-124 targets small C-terminal domain phosphatase 1 (SCP1), associated with the anti-neural repressor element 1 (RE-1)-silencing transcription factor (REST) and the well-characterized splicing factor polypyrimidine tract binding protein (PTBP1/PTB/hnRNP I) [51,59]. Interestingly, REST and PTBP1 regulate expression of miR-124 host genes and processing of miR-124 precursors, respectively. In mouse primary cortical progenitors, REST targets all three miR-124 loci for transcriptional silencing, forming a double negative miR-124-REST/SCP1 feedback loop [60]. In mouse neuroblastoma cell lines, miR-124-mediated repression of PTBP1 initiates a transition to alternative splicing and biogenesis of neuronal specific mRNAs [51]. In 2018, Yeom et al. elucidated a negative feedback loop between these two molecules, whereby PTBP1 binding and blockade of DROSHA/DGCR8-dependent cleavage of pri-miR-124-1 represses miR-124 biogenesis in mouse embryonic stem cells (mESCs) [61]. An additional layer of complexity is the reported competition between PTBP1 and miR-124 for miRNA target sites in the 3′UTR of the *SCP1* gene in HeLa cells [62]. Thus, a secondary source of miR-124 or miRNA targeting either REST, PTBP1, or both, must break up the aforementioned negative feedback loops to allow for neuronal lineage commitment. 

An early study found that in vitro overexpression of miR-124 duplexes and inhibition by anti-miR-124 2′-O-methyl oligonucleotides in murine neural precursors had no significant effects on the neuron/astrocyte ratio (Tuj1^+^/GFAP^+^) during lineage differentiation [63]. However, recent miR-124 deletion studies performed in human and mouse models have reported modest neural lineage commitment impairment. Deletion of all miR-124 encoding alleles in human induced pluripotent stem cells (hiPSCs) and in vivo inhibition of miR-124 in neonatal mouse brains reduced neuronal lineage commitment and neurogenesis, respectively [52,56]. In the mouse model, the authors observed increased gliogenesis in the adult olfactory bulb [56]. One mechanism by which miR-124 orchestrates the fate between neuronal and glial differentiation is through regulation of EZH2 expression, either directly or indirectly through USP14 [64,65]. The discrepancies regarding whether miR-124 alone is essential in neuronal and glial lineage establishment between these studies may be due to the experimental models utilized. 

### 3.2. Convergence of Distinct Pro-Neurogenic miRNAs on Key Inhibitors of Neuronal Differentiation

The modest effects of miR-124 deletion on neuronal lineage development [52,56] suggests that multiple neuronal miRNAs may orchestrate neuronal differentiation by acting in a cooperative fashion to target master regulators of transcriptional and post-transcriptional programs. Theoretically, convergence of miRNAs on key mRNA targets increases the robustness of developmental programs while simultaneously preventing aberrant differentiation. Additionally, coordinated miRNA targeting may help ramify the cellular pool of individual miRNAs amongst hundreds of target mRNAs [66]. Evidence supporting a cooperative pro-neuronal miRNA network has begun to emerge as miR-128, -124 and -137 have been proposed to serve as a triad of pro-neurogenic miRNAs with extensive overlap in key predicted anti-differentiation transcription factor (TF) targets [67]. Analysis of overlap in differentially expressed targets under antagomir treatment of each of the above miRNAs identified specificity protein 1 (SP1), a well-studied transcriptional activator, as a central node at which all three miRNA target networks converge. Supporting this finding, miR-124 has been shown to target *SP1* for downregulation during neurogenesis [68]. Additionally, in non-neuronal systems, miR-128 and miR-137 have been shown to directly target SP1 [69,70,71]. 

Despite the fact that miR-124 deletion alone failed to significantly impact neuron-glia fate in cultured neural precursors [63], miR-124 can promote neuronal lineage establishment in tandem with miR-9/9*. Simultaneous overexpression of miR-9/9* and miR-124 duplexes in neural precursors caused a significant reduction in astrocytic GFAP^+^ cells, thereby increasing the neuron/glia ratio (Tuj1^+^/GFAP^+^) upon differentiation in vitro, compared to control. These effects on cell differentiation were not observed when either duplex was delivered separately [63], highlighting the combinatorial potency of neural miRNAs to advance differentiation. Later, Yoo et al. elucidated a molecular mechanism underlying the pro-neurogenic role of miR-9/9* and -124. Ectopic expression of miR-9/9* and miR-124 directly reprograms primary human dermal adult fibroblasts into functional neurons by targeting a subunit of the Brg/Brm-associated factor (BAF) chromatin remodeling complex, BAF53a, of the neural-progenitor-specific BAF (npBAF53) complex [72], allowing for de-repression of neuron-specific homolog BAF53b.

Analysis of multiple independent miRNA studies establishes the repressor element 1 (RE-1)-silencing transcription factor/neuron-restrictive silencer factor (REST/NRSF) complex as another anti-neural transcriptional hub [73] at which brain-enriched miRNAs seemingly converge. This complex is comprised of the namesake RE1 silencing transcription factor (REST) which forms a complex with co-factors mSin3A and CoREST to bind the 23bp repressor element 1 (RE1) [74,75]. The REST complex recruits histone deacetylases (HDACs) to actively repress transcription of neuronal genes [73]. Neural miR-9 and -9* have been shown to target REST and CoREST, respectively [26]. Additionally, as discussed earlier, miR-124 targets SCP1 [59], which is recruited to genes harboring RE1 elements to silence neuronal genes [76]. In addition to directly targeting components of the REST complex, miR-9/9* and -124 target ubiquitin-specific protease 14 (USP14), which leads to the destabilization and repression of EZH2 and subsequently, destabilization of REST [65]. Thus, miR-9/9* and -124 targeting of proteins stabilizing REST may serve as an additional mechanism to enforce acquisition of neural fate. In addition to regulation of neural-specific transcription networks, neural miRNAs may also converge on a broader spectrum of targets to drive pro-neural posttranscriptional networks, but this paradigm remains largely unexplored.

### 3.3. MiRNAs Govern Glial Lineage Development and the Functional Interplay between Neurons and Glia 

Distinct miRNAs species govern the development and function of various glial cell types in the brain. In oligodendroglia (OL), which are responsible for CNS myelination and providing structural and metabolic support for neurons [77,78,79], miR-219 and miR-338 were among the first group of OL-specific miRNAs identified. These miRNAs are up-regulated 10–100 fold upon differentiation of OL progenitor cells (OPCs) and play crucial roles in advancing OL and myelin development [80,81]. Furthermore, miR-219 directly represses numerous anti-differentiation targets in OLs, represented by platelet-derived growth factor alpha (PDGFRα), Hes5, Sox6, FoxJ3, ZFP238, Lingo1 and Etv5 [80,82]. Notably, miR-219 alone can partially rescue severe OL defects caused by the loss of Dicer, due to its necessary and sufficient roles in proliferation and early differentiation of OPCs, as well as metabolic regulation of lipid formation and myelin assembly [80,82]. More recently, deletion of miR-338 was shown to exacerbate the hypomyelination phenotype caused by miR-219 deficiency, suggesting coordination between miR-219 and miR-338 in promoting CNS myelination and lesion repair [82]. Along with miR-219 and miR-338, a growing number of miRNAs have been shown to participate in OL differentiation and myelin maintenance, as well as in the pathogenesis of demyelination-related diseases, represented by multiple sclerosis, which has been reviewed recently [8]. Among these newly identified pro-myelinating miRNAs, miR-146a was reported to enhance remyelination and axonal protection caused by experimental demyelination and oligodendrogenesis in animal stroke models [83,84]. In contrast to pro-myelination miRNAs, miR-212 inhibits OL maturation by targeting and suppressing mRNAs that drive OL differentiation and myelination, represented by proteolipid protein 1 (PLP1) [85]. 

It should be noted that many well-characterized pro-neurogenic miRNAs were recently detected in OPCs and their decline is crucial for OL development, represented by miR-9 [86]. The expression of another pro-neurogenic miRNA, miR-125a-3p, in OLs prevents myelin cell maturation by targeting cell-cell interactions and Wnt signaling [87]. An additional example of miRNAs differentially affecting distinct neural cell types is the miR-302/367 cluster. Exposure to the histone deacetylase inhibitor valproic acid (VPA) and forced expression of miR-302/367 can reprogram astrocytes into functional neurons or myelinating OLs in specific animal models of neurodegeneration or myelin lesion, respectively [88,89]. These findings highlight the underappreciated fact that one miRNA can distinctly affect brain function when expressed in different neural cell types. Furthermore, an increasing volume of recent studies revealed that miRNAs can directly modulate neuron-glia interactions. For instance, microglia-produced miR-124 can be secreted into small extracellular vesicles (EVs), also referred as exosomes, to enhance neurogenesis and alleviate neurodegeneration [90,91]. On the other hand, miR-124 in neuronal exosomes can mediate neuron to astroglia communication, when it is internalized into astrocytes, leading to up-regulation of the predominant glutamate transporter GLT1 [92], a key player for astrocyte-mediated synaptic plasticity [93]. 

## 4. Neural miRNA Multigene Families

Numerous miRNAs that control neuronal development are derived from multigene families, represented by miR-9, -128, -125, and -124 [94]. In contrast, miR-137 is encoded by a single gene, which harbors numerous genetic alterations that are causatively associated with a number of neuropsychiatric diseases [95,96]. Therefore, miRNA multigene families may represent an evolutionary fail-safe to protect against perturbations/mutations in an essential miRNA gene leading to disease. Additionally, parallel transcription and processing of the same miRNA from different genes allows for robust upregulation and abundant expression of the miRNA, by overcoming the limiting rate of RNAPII transcription of a single gene. Another potential advantage of miRNA multigene families is to allow for different spatiotemporal expression of individual miRNA paralogs in different neuronal cell types or even within the same neuron, possibly through different rates of transcription and/or processing efficiencies. Excitingly, a recent study by Bofill-De Ros et al. reported that pri-miR-9-1 undergoes alternative cleavage by Drosha to generate an isomiR with a unique 5′ end and seed sequence [97]. This finding indicates that a single miRNA paralog within a multigene family can produce two distinct miRNAs from the same pri-miRNA, which may be functionally advantageous in different cellular contexts. 

The exemplary miR-124 multigene family undergoes cell type- and developmental stage-specific regulation of miRNA paralogs to drive neuronal differentiation. Both human and mouse miR-124 are encoded by three paralogous genes, giving rise to pri-miR-124-1, -2 and -3 [55,94]. In the mouse genome, the host transcripts of *miR-124-1* (*Retinal non-coding RNA 3*, *Rncr3*) and *miR-124-2 (Mir124-2 host gene, Mir124-2hg)* have been annotated, while *miR-124-3* is intergenic. In the human genome, only *miR-124-2* (8q12.3) is annotated as an intronic miR of a long-noncoding pri-miRNA (lnc-pri-miR) [98], while *miR-124-1* (8p23.1) and *miR-124-3* (20q13.33) are intergenic. As a multigene family with paralogous miRNAs located in distant regions and/or different chromosomes, the *miR-124* paralogous genes likely arose from a non-local gene duplication event during vertebrate evolution [99]. Despite differences in chromosomal location and genomic position, processing of all three miR-124 paralogs is Drosha, DGCR8 and Dicer dependent [100,101], indicating that each miR-124 paralog undergoes canonical miRNA processing. 

In mouse embryonic stem cells (mESCs), pri-miR-124-1 is the predominantly expressed pri-miR-124 paralog [61]. The expression of pri-miR-124-1 continues to increase as mESCs are differentiated into cortical neurons [61]. Similarly, pre-miR-124-1 is abundantly expressed in the midbrain, frontal-cortex, cerebellum and hippocampus of human adults [102]. Homozygous deletion of the miR-124-1 host transcript (*Rncr3*^−/−^) in mice results in increased neuronal apoptosis, axonal mis-sprouting and an overall smaller brain size [103], demonstrating that miR-124-1 is necessary for proper murine brain development. Similar to pri-miR-124-1, murine pri-miR-124-2 increases from mESCs to cortical neurons [61], which, in tandem with pri-miR-124-1 upregulation, may underlie the robust increase in mature miR-124 observed in differentiating neurons. The role of pri-miR-124-3 in neural lineage has not been characterized as reports show this transcript is negligibly expressed in mESCs and minimally expressed in cortical neurons [61]. Whether similar or distinct regulation of miR-124 paralogs occurs during human neuron development, as compared to rodents, has not been reported. 

## 5. Regulation of miRNAs in Neural Development: Divergence by Biogenesis and Convergence on Key Targets

### 5.1. Transcriptional Regulation of miRNA Genes

#### 5.1.1. Transcriptional Regulation of Neuronal miRNA Genes

As miRNA host genes are primarily transcribed by RNAPII, pri-miRNAs are generally capped and polyadenylated [104] (Figure 1A). Important exceptions are long non-coding transcripts containing miRNAs (lnc-pri-miRNAs), which were found to be predominantly non-polyadenylated (pA^-^) [98]. Processing of pri-miRNAs is coupled to RNAPII transcription, with pri-miRNA processing occurring co-transcriptionally, prior to splicing [105,106,107]. Due to the functional integration of these processes, pri-miRNAs are labile in nature and present at a low abundance at steady-state. Identification of miRNA gene promoters and transcription start sites (TSSs) is challenging and therefore, incomplete.

Ozsolak and colleagues reported that the promoters of RNAPII-transcribed miRNAs share the same features as protein-coding gene promoters, namely a CpG island and eukaryotic core promoter elements including a TATA box, TFIIB recognition element (BRE), initiator (Inr) and downstream promoter element (DPE) [108] (Figure 1A). Of note, these eukaryotic promoter elements are present only in the minority of miRNA promoters, hence a genome-wide analysis of these elements may miss the majority of miRNA genes promoters [108]. Previous studies have systematically characterized miRNA host gene TSSs and promoters by RNAPII ChIP and histone modification ChIP, namely trimethylation of Lys4 of histone H3 (H3K4me3) [109,110]. H3K4me3 modifications deposited around the TSS of miRNA genes are indistinguishable from protein-coding TSSs [108]. Functionally, this modification regulates transcription of protein-coding and miRNA genes by a similar mechanism as reported by genome-wide analysis of ChIP-seq data in mouse ESC and NPCs cells [111]. These studies reveal an intricate regulatory feedback loop between miRNAs and epigenetic machinery when considered in tandem with neural miRNA targeting of epigenetic regulators discussed in Section 3.2.

As the majority of pri-miRNAs harbor a 5′ cap, these transcripts are also candidates for cap analysis of gene expression sequencing (CAGE-seq) [112], which identifies TSSs and promoter regions of RNAPII-transcribed transcripts in a high-throughput manner. CAGE-seq has been used to quantify miRNA expression levels by the proxy measure of pri-miRNA expression [113]. The 5th edition of the FANTOM (Functional Annotation of Mammalian genome) (FANTOM5) project includes CAGE profiling data of pri-miRNA promoter regions and expression in numerous neurodevelopmentally relevant samples, including human iPSC differentiated neurons, human embryonic and adult brain samples, as well as time-course data of the mouse cerebellar miRNA transcriptome from E11 to P9 [113]. For pri-miRNAs encoded by multigene families, these data are particularly useful in identifying cell- and tissue-specific expression among multiple pri-miRNA paralogs, as well as miRNA host transcript promoters for investigation of transcriptional regulation. 

The significance of miRNA host transcript analyses remains dubious as the correlation between expression of miRNA host transcripts and embedded intragenic miRNAs remains unclear. Early human and mouse miRNA studies across different organs revealed frequent coordination between intronic miRNAs and host gene mRNAs [94,114]. More recently, next-generation sequencing revealed a limited correlation between host gene and mature miRNA expression and rather indicated that processing of pri-miRNAs by Microprocessor is a more accurate determinant of miRNA abundance [115]. These inconsistencies may be remedied by accounting for phylogenetic age/evolutionary conservation of miRNAs, as one report found that conserved intragenic miRNAs are co-expressed with their host genes whereas non-conserved miRNAs are uncoordinated with their host genes [116]. Additionally, discrepancies may be, in part, due to incorrect annotation of miRNA host genes [117]. Lastly, another important consideration is the reported use of transcription sites independent from the host gene by one-third of intronic miRNAs [108] (Figure 1A). Future reciprocal analyses of transcriptional and post-transcriptional regulation of miRNA expression will better define the functional determinants of miRNA expression. 

Transcriptional regulation of neuronal miRNAs and the functional impact on brain development has been increasingly recognized. Co-operation of the DNA methyl-CpG-binding protein 2 (MeCP2) and the transcription factor Sox2 in stem cells has been reported to control miR-137 expression in adult neural stem cells, providing an intriguing cross talk between miRNA and chromatin modification-mediated epigenetic regulation in adult neurogenesis [118]. In addition, regulatory feedback loops between a miRNA and transcription factors modulating its expression during neuronal lineage specification have emerged from several independent transcription factor-miRNA focused studies. One example is the double negative miR-124-REST/SCP1 feedback loop, which controls miR-124 expression in mouse neural progenitor cells [59,60]. Another potential double negative feedback loop is formed between miR-9 and REST. Interestingly, all three miR-9 loci are occupied by REST in a murine kidney cell line [60] and miR-9 targets REST in HEK293 cells [26]. Moreover, miR-9 has been shown to participate in double negative feedback loops with Hes1 and TLX to regulate neuronal differentiation [119,120]. 

#### 5.1.2. Transcriptional Regulation of Oligodendroglial miRNA Genes

Glial miRNA host genes are also subjected to transcriptional regulation. One recent report demonstrated that the well-characterized transcription factor Sox10 enhances production of miR-335 and miR-338, two miRNAs known to play critical roles in OL and myelin development [121]. The precursors of miR-338 and miR-335 are located in the introns of the host gene transcripts from *Aatk* and *Mest,* respectively [121]. Sox10 binds the proximal promoter regulatory regions of both host genes, leading to transcriptional activation. Both miR-335 and 338 target the 3′UTR of *Sox9*, which in turn suppresses Sox9 and drives OL terminal differentiation [121].

### 5.2. Posttranscriptional Regulation of miRNA Biogenesis 

In recent years, a growing number of studies have investigated specific pri-miRNA characteristics for Microprocessor substrate recognition and cleavage [122,123,124]. Within the burgeoning epitranscriptomics field, methyltransferase-like 3 (METTL3) was found to catalyze N^6^-methyladenosine (m^6^A) modifications of pri-miRNAs to enhance DGCR8 recognition and efficient processing by the Microprocessor complex [124] (Figure 1A). Importantly, the deposition of m^6^A is not dependent on genomic position as m^6^A modifications were found on both inter- and intragenic pri-miRNAs [124]. 

Several *cis* motifs within pri-miRNAs which influence Microprocessor efficiency have been identified, namely UG at the 5′ basal junction of the stem-loop and UGUG/GUG in the apical loop [123] (Figure 1A). Another form of posttranscriptional regulation is imparted by *trans* acting factors which may bind to pri-miRNAs to modulate cleavage by Microprocessor. Some *trans* acting co-factors, which enhance processing of specific pri-miRNAs, include SRp20 (SRSF3), QKI, p72 (DDX17), KSRP, hnRNP A1, BRCA1, FUS and SF2/ASF [123,125,126,127,128,129,130,131,132,133] (Figure 1B). These RBP co-factors have been reported to promote processing of pri-miRNAs by interaction with and recruitment of Microprocessor to the local stem-loop [125,127,130,131] and inducing a relaxed conformational change in the stem-loop, favorable for Microprocessor cleavage [129]. 

Conversely, known *trans* acting co-factors which repress processing of specific pri-miRNAs include PTBP1, NF90-NF45, MSI2/HuR, hnRNP A1 and Lin28B [61,134,135,136,137] (Figure 1B). Mechanisms by which these RBPs negatively affect miRNA processing are similar to those employed by RBPs which promote processing, including binding proximally or directly to the stem-loop, effectively blocking Microprocessor access to the cleavage site [61,134]. Additionally, these RBPs can induce a rigid conformational change in the stem-loop, which is unfavorable for Microprocessor cleavage [136] and compete with RBPs which promote processing for binding to shared pri-miRNA targets [135]. Therefore, variations in pri-miRNA sequence composition between miRNA family paralogs and species, combined with differential RBP abundance in specific cell-types and systems, allow for differential pri-miRNA processing and expression of the derivative miRNAs [115]. 

During brain development, lineage-specific and/or developmental stage-specific expression has been reported for some RBPs mentioned here. Notably, QKI expression has been detected in neural stem/progenitor cells but is selectively silenced in maturing/differentiating neurons [138]. On the contrary, QKI is markedly upregulated during oligodendroglia and myelin development [139,140]. Another example is the decreased expression of DDX17 during in vitro differentiation of human SH-SY5Y neuroblastoma cells [141]. Additionally, PTBP1 is abundantly expressed in NPCs and translationally repressed by miR-124 during neuronal differentiation [51]. At this moment, how developmental regulation of these RBPs impacts processing of numerous pri-miRNA targets to affect brain maturation remains to be determined. 

### 5.3. Posttranscriptional Regulation of miRNA Activity

Circular RNAs (circRNAs) are single-stranded, closed loop RNA structures, produced by non-canonical ‘back-splicing’ that covalently links a downstream splice donor site to an upstream splice acceptor site [142,143]. These RNA molecules are highly expressed in the brain. Notably, the human brain expresses significantly more circRNA species than rodent brains in various brain regions, including the cortex, cerebellum and hippocampus [144]. Increasing evidence suggests functional roles for circRNAs as miRNA sponges [145,146]. Indeed, knockout of neuron-specific circRNA CDR1 antisense (CDR1as) in mice, which harbors multiple conserved miR-7 binding sites, led to a brain-specific decrease in miR-7, upregulation of miR-7 targeted immediate early genes (IEGs), and subsequent dysfunction of excitatory synaptic transmission [147]. In a broader context, altered circRNA expression has been documented in post-mortem brains of schizophrenia (SCZ) and Alzheimer’s disease patients [148,149]. Perturbations in circRNA-mediated sponging of disease-relevant miRNAs have been proposed to underlie disease pathogenesis yet further investigations are needed to reach definitive conclusions [148,149]. 

A second class of non-coding RNAs with the potential to regulate miRNA function/abundance are long noncoding RNAs (lncRNAs). Defined as RNAs more than 200 nucleotides in length without protein-coding capacity, lncRNAs are highly expressed in the brain with known roles in transcriptional and posttranscriptional regulation of coding genes [150,151,152]. Currently, there are four paradigms of lncRNA-miRNA interactions: (1) lncRNAs sequestering/sponging miRNAs, (2) miRNAs affecting lncRNA stability, (3) lncRNA-miRNA competition for mutual target mRNAs and (4) lncRNAs as miRNA precursors [153]. Precise mechanisms and functional importance of lncRNA-miRNA interaction paradigms are under active investigation. 

## 6. Neural miRNAs in Brain Disorders: Dysregulation, Etiology, and Therapeutic Potential

### 6.1. Neuronal miRNAs in Brain Diseases

Abnormalities in neuronal development underlie the pathogenesis of numerous neuropsychiatric diseases, including schizophrenia, major depression, autism and fragile X intellectual disability [154,155,156]. Given the spatiotemporal function of miRNAs in governing normal neuronal development, genetic alterations and dysregulations that result in abnormalities of numerous miRNAs have been reported in various brain disorders, which are well documented in recent reviews [11,12,157]. Well-characterized examples include miR-137 in major mental illnesses [95], miR-125a in fragile X intellectual disability [158], and miR-124 in neuropsychiatric, neurological and neurodegenerative disorders [157,159,160].

Beyond neurodevelopmental disorders, the role of embryonic and adult neurogenesis in neurodegenerative disorders is increasingly recognized. In Parkinson’s disease (PD), malfunction of PD-related genes affects neural stem cell proliferation and maintenance [161]. Thus, abnormalities in early neuronal development not only underlie neurodevelopmental disorders but may also contribute to age-related neurodegenerative disorders. Besides the well documented miRNA abnormalities identified in post-mortem brains of neurodegenerative disorder patients [162,163,164], the rapid advancement in iPSC technology has enabled identification of mechanisms of miRNA dysregulation in iPSC-derived neurons of various neurodegenerative disease patients, such as amyotrophic lateral sclerosis (ALS) and PD [15,165]. Moreover, the therapeutic potential of miRNAs in stem cell-based treatments for neurodegenerative diseases has gained significant attraction in basic and clinical neuroscience. In particular, miR-9/9* and miR-124 have been shown to generate disease-relevant neuronal subtypes, including striatal medium spiny neurons, cortical neurons, and spinal cord motor neurons, providing promise for miRNA-based stem-cell therapy [166]. 

Perturbations in miR-124 are implicated in several neuropsychiatric and neurodegenerative diseases, including schizophrenia, Alzheimer’s disease (AD), PD, hypoxic-ischemic encephalopathy, Huntington’s disease, and ischemic stroke [167]. The human miR-124-1 locus is located in chromosome 8p23.1, which harbors genes that have been implicated in schizophrenia, microcephaly and epilepsy [168,169,170,171]. Increased expression of mature miR-124 has been found in the hippocampus of AD patients and the hippocampus of Tg2576 mice, a model of AD, as well as in the prefrontal cortex of major depressive disorder (MDD) patients [172,173]. However, other studies have reported decreased expression of mature miR-124 in the frontal cortex of sporadic AD patients [174]. Mechanistically, miR-124 is reported to target and repress beta-site amyloid precursor protein cleaving enzyme 1 (BACE1) [174,175], an enzyme that cleaves amyloid precursor protein (APP) to β-Amyloid (Aβ). Therefore, in AD pathology, downregulation of miR-124 may indirectly lead to accumulation of Aβ and subsequently, amyloid plaque formation. In this model, miR-124 is reported to be neuroprotective [175]. However, it is important to measure differences in miR-124 across all major brain regions in AD patients compared to controls before more definitive conclusions can be made.

*Rncr3*^−/−^ mice, homozygous for deletion of the miR-124-1 host transcript, exhibit abnormal front and hind limb clasping, commonly observed in mouse models of neurodegenerative disorders [103]. In line with the disease phenotype observed in *Rncr3^−/−^* mice, decreased expression of mature miR-124 has been reported in a mouse model of and in the frontal cortex of subjects with behavioral variant frontotemporal dementia (bvFTD) [176]. Decreased expression of mature miR-124 has also been measured in the cortex of R6/2 mice, a model of HD but importantly, there was no significant dysregulation of miR-124 in the cortex of HD patients as compared to controls [177]. In human and mouse models of PD, miR-124 has been proposed to serve as a neuroprotective molecule by targeting numerous different disease-relevant targets. In a mouse model of PD, miR-124 targets p38 and p62 to inhibit activation of the microglial inflammatory response in PD [178]. In human PD cell models, miR-124 targets signal transducer and activator of transcription 3 (STAT3) and phosphorylated 5′ adenosine monophosphate-activated protein kinase (p-AMPK) of the AMPK/mTOR pathway, which play roles in microglial activation and cell apoptosis and autophagy, respectively [179,180]. Moreover, reduced miR-124 has been proposed as a diagnostic biomarker for PD and the therapeutic potential of miR-124 in the treatment of PD has been recognized [159]. A comprehensive review of the therapeutic potential of miR-124 in other neurodegenerative disorders can be found elsewhere [181]. Taken together, these human and mouse data indicate that precise expression of miR-124 is crucial for proper brain development and functioning, with disease phenotypes arising under elevated and decreased miR-124 conditions.

MiR-137 is another brain-enriched miRNA that governs differentiation and maturation of the nervous system and is implicated in various major mental illnesses [182]. Aberrant increase and decrease of miR-137 expression were reported in post-mortem studies of intellectual disability, autism spectrum disorders (ASD), and schizophrenia (SCZ) [95,96,183]. Moreover, multiple genetic alterations in human miR-137 variant alleles are found highly associated with SCZ and other neuropsychiatric disorders in the largest mega genome-wide association studies [95]. Recently, a variable number tandem repeat (VNTR) element in the *MIR137 Host Gene* was found to affect alternative splicing and biogenesis of miR-137, which contributes to the risk for SCZ [184]. In genetically engineered mouse models, homozygous loss of miR-137 in the germline or conditional knockout (cKO) in the nervous system leads to postnatal lethality, while viable heterozygous cKO mice display repetitive behavior, impaired sociability and learning [185]. Mechanistically, miR-137 regulates numerous targets that play crucial roles in governing neuronal maturation and neoplastic transformation [182]. In addition, miR-137 has been shown to regulate several forms of synaptic plasticity that are proposed to underlie SCZ pathogenesis [186,187]. Moreover, aberrant up-regulation of the phosphodiesterase 10a (Pde10a), a direct miR-137 target, was found to underlie the behavioral abnormalities caused by partial loss of miR-137 [185]. 

### 6.2. MiRNAs in Glial Cells and Glia-Neuron Communication Implicated in Brain Diseases

It is increasingly appreciated that defects in both neurons and glia underlie the pathogenesis of brain diseases. Deficits in myelinogenesis and neuronal function contribute to distinct aspects of the complex etiology of SCZ [188,189]. Interestingly, association between single nucleotide polymorphisms in miR-219-1 and miR-137, which are crucial for oligodendroglia and neuronal lineage development, respectively, has been reported in a SCZ cohort [190]. This finding suggests that disruptions in glial miRNAs and neuronal miRNAs may synergistically manifest the same brain disease. 

Besides myelin–neuron interactions, communication between microglia and astrocytes with neurons modulates neuronal cell death, neurogenesis and synaptic interactions during normal brain development, as well as neuroimmune-responses in various brain diseases [191,192,193]. In particular, secreted extracellular vesicles (EVs) are key players in neuron-glia intercellular signaling and have recently been implicated in the pathogenesis of neurodegenerative disorders [193,194,195]. Distinct profiles of EV miRNAs, derived from neurons and glia, are found in various CNS diseases, often before the onset of irreversible neurological damage. As EVs can pass from brain into the blood, these molecules could serve as promising biomarkers [196,197]. A recent report revealed that miR-23a-3p, miR-126-3p, and miR-151a-3p are significantly reduced in brain-derived plasma EVs from AD patients compared to controls [198]. In a separate cohort, the levels of miR-126-3p, miR-142-3p, miR-146a-5p, and miR-223-3p in brain-derived EVs were reported to correlate with disease severity of AD [199]. Besides neuronal EVs, astrocyte-secreted EVs also contain unique miRNAs. Upon exposure to pro-inflammatory cytokines, astrocyte-secreted EVs are enriched in miR-125a-5p and miR-16-5p, which are thought to exacerbate the pathology of neurodegenerative diseases such as AD, PD, ALS, and stroke [194]. Additionally, inflammatory microglia shed EVs enriched in the microglia-specific miR-146a-5p, which is transferred to neurons to suppress presynaptic synaptotagmin1 [200].

Finally, exciting discoveries suggest therapeutic potentials of EV miRNAs derived from neural cells. NPCs are known to have potent therapeutic effects in neurological disorders through EV secretion. In a recent report, miR-21a was found to be highly enriched in NPC-originated EVs, which promoted neuronal differentiation but not gliogenesis [201]. In addition, mesenchymal stromal/stem cells secreted EVs are enriched in miR-467f and miR-466q, which modulate the pro-inflammatory phenotype of microglia acutely isolated from late symptomatic SOD1(G93A) mice, a murine ALS model [202]. Besides naturally secreted EVs by brain cells, specific miRNAs, represented by miR-124, can be loaded into engineered EVs, which has been shown to attenuate cocaine-mediated microglia activation [203] and radiation-induced brain injury [204]. 

The miRNA species mentioned in this review, the brain cell types in which they are expressed, their roles in neural cell development, and their relevance to various brain disease are summarized in Table 1.

## 7. Concluding Remarks and Perspectives

In comparison to the well-documented mechanisms of miRNA action and the expression profiles of brain miRNAs produced by the canonical miRNA biogenesis pathway, our current understanding of the molecular mechanisms that regulate miRNA expression during neural development and miRNA dysregulation in various human brain disorders are still in their infancy. The new advancements regarding the multifaceted regulation of neuronal and glial miRNAs in healthy and diseased brains, reviewed here, highlight the importance of further delineating fundamental rules that govern miRNA production and activity in normal and diseased brains. The complex regulation of miRNAs provides spatial and temporal divergence of miRNA-dependent gene regulation to accommodate sophisticated brain function as well as fragility to human brain diseases caused by miRNA dysregulation. By contrast, the emerging examples of lncRNAs/circRNAs in soaking multiple miRNAs derived from distinct genes may provide a mechanism for functional re-organization of miRNA activity that converge on downstream targets, especially in neuronal somatodendritic compartments. Nonetheless, the current prevailing challenge is delineating how miRNAs converge on a vast number of downstream mRNA targets to advance neuronal and glial differentiation, especially considering the emerging roles of miRNAs in neuron–glia communication. Finally, therapeutic potentials of pro-neurogenic miRNAs have become an intriguing area of active research that may provide promising outcomes. 

## Figures and Tables

**Figure 1 ijms-22-06765-f001:**
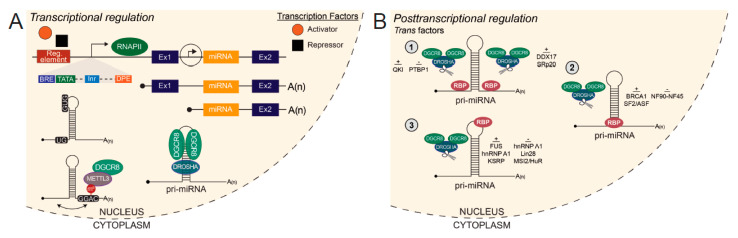
Transcriptional and posttranscriptional regulation of miRNA biogenesis. (**A**) RNA polymerase II (RNAPII)-mediated transcription of a miRNA host gene from an upstream host gene promoter or independent transcription start site produces two variant pri-miRNAs. Transcription factors bind to transcriptional regulatory elements proximal to the transcription start site (TSS) to activate or repress gene expression. At the TSS, miRNA host gene promoters harbor the same features as eukaryotic protein-coding gene promoters. *Cis* elements, UG at the 5′ basal junction and UGUG/GUG in the apical loop, discriminate pri-miRNAs from other secondary structures in the transcriptome and recruit Microprocessor for cleavage. The methyltransferase METTL3 deposits m6A modifications on pri-miRNAs to enhance recognition by DiGeorge Syndrome Critical Region 8 (DGCR8). Drosha and DGCR8 bind the pri-miRNA as a heterotrimeric complex to cleave and generate a pre-miRNA for further downstream processing. (**B**) *Trans* acting factors known to bind pri-miRNAs that positively or negatively regulate Microprocessor cleavage. (1) Factors which bind to the single-stranded RNA flanking the stem-loop. Binding toward the 5′ or 3′ end of the pri-miRNA is specified. (2) Factors which bind to the stem-loop. (3) Factors which bind to the apical loop. Regulatory element, Reg. element; TFIIB recognition element, BRE; TATA box, TATA; Initiator, Inr; Downstream promoter element, DPE; RNA Polymerase II, RNAPII; Exon, Ex; Methyltransferase-like 3, METTL3.

**Table 1 ijms-22-06765-t001:** Neuronal and glial miRNAs in brain disease. References in the table correspond to references in the body of the text of this review article.

Cell Type/System	MicroRNA	Function in Brain Development	Brain Disease Relevance
Neuron	miR-124	Promotes neuronal differentiation [50,51,52,53], neuron-glia communication	AD [173,174,175], HD [177], MDD [172], Frontotemporal dementia [176]
	miR-137	Neuronal differentiation, synaptic activity	SCZ [95], ASD [183], Intellectual disability [96]
	miR-9	Promotes neuronal differentiation [27]	HD [26]
	miR-7	Immediate early genes, synpatic transmission [147]	
	miR-125a	Neuronal translation, synaptic function	Fragile X [158]
Microglia	miR-124	Inhibits microglial activation	PD [178,179,180]
	miR-146a-5p	Modulates synpase function [200]	Correlated with AD severity [199]
Oligodendroglia	miR-219	OL development, myelination and repair	SCZ [189], Multiple sclerosis [8]
	miR-338	OL development, myelination and repair	Multiple sclerosis [8]
	miR-146a	Myelination and repair	Multiple sclerosis [8]
	miR-212	Prevents OL maturation [85]	Myelin disorders
	miR-125a	Prevents OL maturation [87]	Myelin disorders
	miR-335	OL development [121]	Myelin disorders
Astroglia	miR-451		Glioma tumorigenesis [48,49]
	miR-125a	Extracellur vessicle	AD, PD, ALS, and stroke [194]
	miR-16-5p	Extracellur vessicle	AD, PD, ALS, and stroke [194]
	miR-302/367	Reprogram astrocytes to neuronal or OL fate [88,89]	Potential repair of neurodegeneration or myelin lesion
Brain exosomal miRNAs	miR-21a	Promotes neuronal differentiation [201]	
	miR-467f	Pro-inflammatory phenotype of microglia	ALS [202]
	miR-466q	Pro-inflammatory phenotype of microglia	ALS [202]
	miR-23a-3p	Potential biomarker for AD	Reduced in AD [198]
	miR-126-3p	Potential biomarker for AD	Correlated with AD severity [199]
	miR-151a-3p	Potential biomarker for AD	Reduced in AD [198]
	miR-142-3p	Potential biomarker for AD	Correlated with AD severity [199]
	miR-223-3p	Potential biomarker for AD	Correlated with AD severity [199]

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
