# Peer review of "Multifaceted Regulation of MicroRNA Biogenesis: Essential Roles and Functional Integration in Neuronal and Glial Development"

_ijms, 2021, doi:10.3390/ijms22136765_

Round 1
Reviewer 1 Report
1. This manuscript is well organized but still needs a complete proofreading in English grammar and some spellings.
2. The format of the manuscript can be improved for better understanding. The mechanism of miRNA synthesis and regulation can be shown by figure, while the miRNA functions in diseases can be listed by table.
3. The glia is a complicated conception. They may include microlgia, astrocyte, and oligodendrocyte. The author should make it clear in Introduction, and may discuss miRNA function by cell types.
4. Briefly, this manuscript format should be revised before acceptance for publication. First, the knowledge is complicated so should be illustrated in tables and figures. Secondly, cell type difference should be mentioned and discussed one by one in detail.
Author Response
We appreciate the constructive suggestions by this reviewer and have extensively revised the review manuscript accordingly. Point-to-point responses are provided below.
Reviewer 1
- This manuscript is well organized but still needs a complete proofreading in English grammar and some spellings.
We made sincere efforts to proofread the entire manuscript, corrected errors in English grammar and some spellings.
- The format of the manuscript can be improved for better understanding. The mechanism of miRNA synthesis and regulation can be shown by figure, while the miRNA functions in diseases can be listed by table.
The mechanism of miRNA synthesis and regulation is shown in the original Figure 1, which illustrates transcription and posttranscriptional regulation of miRNA biogenesis. According to the reviewer’s suggestion, we made a new table (Table 1), in which we listed miRNA species mentioned in this review based on the brain cell types they are expressed, their roles in neural cell development, and their relevance to various brain diseases.
- The glia is a complicated conception. They may include microlgia, astrocyte, and oligodendrocyte. The author should make it clear in Introduction and may discuss miRNA function by cell types.
Thanks for this suggestion. We included the subtypes of glia in the introduction. We also listed miRNAs based on the neural cell types they are expressed, their function, and disease relevance in Tab 1.
- Briefly, this manuscript format should be revised before acceptance for publication. First, the knowledge is complicated so should be illustrated in tables and figures. Secondly, cell type difference should be mentioned and discussed one by one in detail.
We included clarifications of glia types in the introduction. We also included a new Table (Tab 1) to list the cell type origin, function, and implication in brain diseases of neuronal and glial miRNAs reviewed in this article. A more complete list of miRNA species in neuronal and synaptic function in relation to brain diseases are provided in several recently published reviews thus not repeated here.
Because this review focuses on the new advancements in regulation of miRNA biogenesis, we organized the article primarily based on the regulation steps and mechanisms that control miRNA biogenesis in neural development. According to the reviewer’s suggestion, we provided subtitles to highlight subsections of neuronal and glial miRNAs within the sections when new advancements are reviewed for both neurons and glia (section 2.1/2.3, section 4.1.1/4.1.2, section 5.1/5.2).
Reviewer 2 Report
Comments and suggestions for authors
The manuscript of Suster and Feng is well-written and structured. The authors focused on the regulation of microRNA biogenesis and their function in neurogenesis, neurodevelopment, and associated disorders.
The followings are my comments to be considered.
- The authors should describe what CNS stands for at the beginning of the text at line 100?
- The authors mentioned that the miR-9 biogenesis is regulated by miR-107-DICER interaction in differentiating zebrafish neuronal cells. How does miR-107 maintain its own biogenesis when it downregulates levels of DICER, an RNase III that generates the mature miRNAs from pre-miRNAs? Can authors comment on it (at lines 89-91)?
- In the manuscript, the authors used miR-124-3p/5p. For better understanding, authors can define the meaning of “3p/5p,” etc., after the name of any sp. miRNA.
- The authors discussed miR-124 that is one of the miRNAs being dysregulated in many neurodegenerative disorders, including Alzheimer’s disease (AD), Parkinson's disease (PD), and Huntington's disease (HD). The authors summarize that miR-124 is increased in AD patients while it is reduced in PD. Can they describe if miR-124 neuroprotective or neurotoxic, by which miR-124 regulates specific target mRNAs and the relevant pathways in AD/PD? For example, miR-124 is implicated in 5' adenosine monophosphate-activated protein kinase (AMPK), the signal transducer and activator of transcription 3 (STAT3), extracellular signal-regulated kinase (ERK), Beta-site amyloid precursor protein cleaving enzyme 1 (BACE1), p62/p38-mediated pathway that coordinates with mitochondrial function, autophagy/mitophagy and apoptosis in AD/PD (PMID: 31707035; PMID: 28867212 and PMID: 30995872).
- It might be better to comment and briefly discuss the role of miR-124 in Huntington's disease (HD) pathology.
Author Response
We were pleased by the reviewer's positive comments “The manuscript of Suster and Feng is well-written and structured. The authors focused on the regulation of microRNA biogenesis and their function in neurogenesis, neurodevelopment, and associated disorders.” We especially appreciate the constructive suggestions and have extensively revised the review manuscript accordingly. Point-to-point responses are provided below.
Reviewer 2
- The authors should describe what CNS stands for at the beginning of the text at line 100?
Thanks for the comment and we provided this in the text as suggested.
- The authors mentioned that the miR-9 biogenesis is regulated by miR-107-DICER interaction in differentiating zebrafish neuronal cells. How does miR-107 maintain its own biogenesis when it downregulates levels of DICER, an RNase III that generates the mature miRNAs from pre-miRNAs? Can authors comment on it (at lines 89-91)?
We clarified this confusion. miR-107 and Dicer forms a negative feedback loop, which governs optimal and balanced pro-neurogenic miRNA production. The text is revised on Page 3.
- In the manuscript, the authors used miR-124-3p/5p. For better understanding, authors can define the meaning of “3p/5p,” etc., after the name of any sp. miRNA.
Following this suggestion, we modified the text on Page 2 to provide the definition of miR-3p and miR-5p, using miR-124 as an example.
- The authors discussed miR-124 that is one of the miRNAs being dysregulated in many neurodegenerative disorders, including Alzheimer’s disease (AD), Parkinson's disease (PD), and Huntington's disease (HD). The authors summarize that miR-124 is increased in AD patients while it is reduced in PD. Can they describe if miR-124 neuroprotective or neurotoxic, by which miR-124 regulates specific target mRNAs and the relevant pathways in AD/PD? For example, miR-124 is implicated in 5' adenosine monophosphate-activated protein kinase (AMPK), the signal transducer and activator of transcription 3 (STAT3), extracellular signal-regulated kinase (ERK), Beta-site amyloid precursor protein cleaving enzyme 1(BACE1), p62/p38-mediated pathway that coordinates with mitochondrial function, autophagy/mitophagy and apoptosis in AD/PD (PMID: 31707035; PMID: 28867212 and PMID:30995872).
We added discussion of alterations of miR-124 expression as well as some of the targets affected in AD and PD (Page 12 and 13).
- It might be better to comment and briefly discuss the role of miR-124 in Huntington's disease (HD) pathology.
We added discussion of the potential roles of miR-124 in a mouse model of HD (Page 12). We also referenced a postmortem study that did not detect dysregulation of mioR-124 in the cortex of HD patients (Page 12).
Round 2
Reviewer 1 Report
The revised figure and table can not be found in the manuscript. I can not provide suggestions.
Author Response
We apologize for the confusion. We uploaded the PDF files of Figure 1 and Table 1 during last submission, and do not know why they not not show up. We now just pasted them into the end of the word file.
Round 3
Reviewer 1 Report
The manuscript were improved and can be accepted for publication.